# Evaluation of Soldiers’ Knowledge and Sense of Threats Regarding Exposure to Biological Risk Factors at the Place of Service

**DOI:** 10.3390/healthcare12171777

**Published:** 2024-09-05

**Authors:** Magdalena Zawadzka, Aleksandra Lis, Justyna Marszałkowska-Jakubik, Paweł Szymański

**Affiliations:** 1Department of Epidemiology and Public Health, Medical University of Lodz, Żeligowskiego 7/9, 90-752 Lodz, Poland; 2Department of Organization of the Health Care System, Prevention and Treatment Team, Military Institute of Hygiene and Epidemiology, Kozielska 4, 01-163 Warsaw, Poland; justyna.jakubik@wihe.pl; 3Department of Pharmaceutical Chemistry, Drug Analyses and Radiopharmacy, Medical University of Lodz, Muszyńskiego 1, 90-151 Łódź, Poland; aleksandra.lis@umed.lodz.pl; 4Department of Radiobiology and Radiation Protection, Military Institute of Hygiene and Epidemiology, Kozielska 4, 01-163 Warsaw, Poland

**Keywords:** biological risk factors, soldiers, military, level of knowledge, exposure

## Abstract

Exposure to harmful biological agents and the level of knowledge about specific risk factors are extremely important topics, especially among military personnel. This study evaluates the knowledge and perceptions of soldiers regarding exposure to biological risk factors during their service. This research was conducted using an online survey distributed through the WBBS research panel, in which 1331 soldiers from various demographic groups and ranks participated. The survey assessed awareness of biological threats, the level of knowledge about preventive measures, and the perceived adequacy of occupational health and safety training. The findings reveal that over 80% of respondents possess some level of knowledge about the types of biological agents they may encounter, yet significant gaps remain, particularly in training related to region-specific infectious diseases, with more than 75% of participants having not received such training. Additionally, approximately 5% of respondents reported high exposure, and around 4% reported very high exposure to harmful biological agents, highlighting the need for enhanced educational programs and preventive measures in military contexts. The study underscores the importance of continuous education and training to mitigate risks associated with biological hazards in military environments.

## 1. Introduction

Soldiers represent a professional group performing their duties in variable and often unpredictable environmental conditions, which are frequently harmful to health and burdensome. The main hazards during soldiers’ service include prolonged field training, extreme exercise, and intensive training, during which soldiers are exposed to various harmful factors (such as physical, chemical, or biological risk factors). Additionally, there is a risk of biological factors, which includes challenges related to infectious diseases during wartime operations, including pathogens endemic to the geographic area of operations, participation in missions in various hygienic and climatic conditions, and wound infections caused by common environmental microorganisms. 

There are two fundamental categories of harmful biological agents: unintentional, such as exposure to pathogens through mosquito bites or tick-borne diseases, and intentional, such as those used in warfare, including exposure to biological weapons [1]. The ease of production, wide availability of biological agents, and technical knowledge, combined with the fact that biological weapons can be more effective than conventional and chemical weapons, have led to the spread of biological risk factors and the associated risks [2].

Soldiers may be exposed to biological risk factors in various situations related to their military service, including operations in regions with a high risk of infectious diseases, such as tropical jungles or areas of armed conflict; field conditions, where soldiers remain in close contact, facilitating the rapid spread of pathogens; exposure to biological weapons, particularly in conflict situations where they may be deliberately used to cause disease; exposure to disease vectors; contamination of the operational environment, often deliberate, including biological, chemical, and radiological agents; and exposure to smoke and toxic fumes [3,4,5,6,7].

Methods of protection against biological risk factors include the use of personal protective equipment, medical prophylaxis, early warning systems for biological threats and rapid response procedures, as well as training and education, which encompass raising awareness of biological threats and teaching the correct use of personal protective equipment [1,8].

In the context of military service, these factors can be particularly significant due to the specific conditions in which soldiers perform their duties. The military is a unique group where the risk of exposure to biological factors is increased, and pathogen transmission occurs on a large scale, including person-to-person transmission, contaminated food and water, and airborne and vector-borne routes. Fieldwork, high-risk regions for infectious diseases, close contact with others, often unhygienic sanitary conditions, stressful work under pressure, and limited access to health services and medicines are the most common mechanisms of pathogen transmission between soldiers and civilians [3]. An additional concern is the potential for biological threats encountered by soldiers to spread beyond military personnel to the civilian population. This risk is heightened in situations where soldiers return from regions with high incidences of infectious diseases or where they have been exposed to biological agents. Pathogen transmission can occur through direct contact, shared resources, or contaminated environments, thereby posing a significant public health risk. The movement of military personnel between different geographic locations and their interactions with civilian communities can facilitate the spread of infectious diseases, underscoring the need for stringent biosecurity measures and public health protocols to mitigate the risk of broader outbreaks.

Therefore, understanding these threats and assessing the level of knowledge and sense of threat among soldiers is essential to developing effective protection strategies, minimizing risk, and ensuring the effective protection of military personnel.

## 2. Materials and Methods

The study was conducted on a nationwide sample of soldiers. The Military Center for Civic Education—Military Bureau of Social Research (WBBS) was responsible for drawing the sample and its implementation. The research tool was an online survey (CAWI). It consisted of 38 substantive questions and 10 socio-demographic questions. The questionnaire used a subjective, individual assessment of the soldiers’ levels of exposure (5-point scale), knowledge (3-point scale), and self-awareness (4-point scale). Participation in the study was voluntary. Respondents were reached through the WBBS research panel. A total of 1331 people took part in the study. After completing the study, WBBS provided the raw test results in EXCEL format.

Consent to conduct the study was obtained from the Ethics Committee. Name: Bioethics Committee at the Military Medical Chamber in Warsaw; Approval code: 11/23; Approval date: 19 May 2023

The aim of the study was to assess respondents’ awareness of exposure and level of knowledge regarding harmful biological factors and preventive measures in this area.

It was hypothesized that the level of knowledge and self-awareness of soldiers regarding exposure to harmful biological factors depends on socio-demographic variables.

### 2.1. Software and Statistical Analysis

The GNU PSPP 2.0.0 program was used to analyze the results. The procedures used in the analysis included:Frequency analysis (single-variable analysis);Correlation analysis (using contingency tables and statistics);Correlation analysis using Spearman’s coefficient.

A 95% confidence interval was adopted for statistical analysis and the significance level was *p* = 0.05, which indicates statistically significant results for levels *p* < 0.05.

### 2.2. Characteristics of the Respondents

All participants in the study were members of the Polish military forces. Among the 1331 respondents, 60 did not report their age, resulting in a sample size of 1271 for age-related analysis. The majority of respondents were men, comprising approximately 94.3% of the sample, while the remaining respondents were women. A significant proportion of the respondents were soldiers aged 31 to 40 years (46.7%); soldiers aged 41 to 50 years accounted for 40.8% of the sample; and 6.7% of respondents were under 30 years old. The smallest group, representing just over 5.9%, were soldiers over 50 years of age. The mean age of the respondents was 40.06 years, with 6.53% of the respondents being 35 years old, making this the most common age group. Additionally, 24.9% of respondents reported living in rural areas, while the remaining 75.1% resided in urban areas of various sizes. Of those living in cities, 43.1% lived in cities with populations up to 50,000; 21.3% in cities with 50,000 to 150,000 inhabitants; 16.7% in cities with 150,000 to 500,000 inhabitants; and 18.9% in cities with over 500,000 inhabitants. Information on place of residence was not provided by 39 respondents.

The respondents exhibited varying levels of education. Only slightly more than 2.0% of survey participants had basic vocational education, 53.5% had secondary education, and 44.2% possessed higher education.

Regarding military rank, 24.9% of the respondents were professional privates, 53.4% were non-commissioned officers, 9.3% were junior officers, and 12.4% were senior officers. The professional experience of the respondents ranged from 3 to 40 years of service. The largest group, comprising 11.3% of respondents, had 15 years of professional experience, with the average length of service being 16.8 years. A detailed analysis of the data revealed that the dominant group (48.9%) consisted of individuals with 11 to 20 years of service experience. Fifty respondents did not provide this information.

In terms of positions held, 19.3% of the respondents occupied staff positions, 21.9% held command positions, 28% were in specialist roles, 15.3% were in other specialist roles, and 15.2% described their positions as “other”. More than half of the respondents (59.4%) had not traveled abroad or participated in missions as part of their official duties, while the remaining 40.6% had engaged in such activities. Among those who had participated in trips and missions, 39.6% were men, 46.7% were aged between 51 and 60 years, 53.3% had vocational education, 56.9% were senior officers, and 54.8% had between 31 and 40 years of service experience. Thirty-seven respondents did not provide information regarding their service positions.

## 3. Results

### 3.1. Biological Agent Risk

Almost half of the respondents are not exposed to harmful biological agents. More than one in four of the respondents work in low exposure and more than one in eight work in medium exposure to the above-mentioned factors. Approximately 5% of the respondents indicated high exposure, whereas approximately 4% indicated very high exposure to harmful biological agents. Detailed data analysis connected with the level of exposure to harmful biological agents depending on gender, age, education, professional corps of military service, length of military service, and official position of the respondents is presented in Table 1.

Over 80% of the respondents possess knowledge in terms of types of biological factors they are exposed to during time of service. One in four study participants have detailed information in this field, one in four have partial knowledge, and one in eight lack knowledge of the subject. Detailed data analysis connected with the level of knowledge regarding the types of biological agents occurring during service depending on gender, age, education, professional corps of military service, length of military service and official position of the respondents is presented in Table 2.

### 3.2. Preventive Actions

More than half of respondents participated in training on the health effects of exposure to harmful factors and prevention against them, and every third person had no knowledge of them. Almost the same number of the respondents believe that training in this subject did not take place and are mindful of their organization. Detailed analysis of the respondents’ consciousness regarding training arranged in the workplace in the area of health effects of exposure to harmful factors and methods of preventing them depending on gender, age, education, professional military service corps, length of military service and the respondents’ official position is presented in Table 3.

Before being assigned to service, more than 75% of people had not taken part in training in the scope of infectious diseases specific to a region concerned (occurrence, prevention, treatment). Detailed analysis of the respondents’ participation in training in the scope of infectious diseases specific to a region concerned depending on gender, age, education, professional corps of military service, length of military service and the respondents’ official position is presented in Table 4.

The feeling of being properly informed about occupational health and safety rules at the workplace are declared by almost all the respondents. Approximately 4.5% of the respondents gave feedback that they could be more educated in this field. Detailed analysis regarding the respondents’ feelings of being properly informed about occupational health and safety rules at the workplace depending on gender, age, education, professional military service corps, length of military service, and the respondents’ official position is presented in Table 5.

## 4. Discussion

The primary objective of this study was to assess the level of exposure to harmful biological agents among soldiers and to analyze how this level varies according to key sociodemographic variables. Proper assessment of this threat is crucial for planning healthcare delivery for military populations, assessing health risks during military deployments, and designing effective vector control strategies [9]. Our results indicate that nearly half of the respondents are not exposed to harmful biological agents, while approximately 5% reported a high level of exposure. The analysis revealed that the level of exposure varies depending on gender, age, education, and military rank, with statistically significant differences observed in each of these categories. For example, older soldiers and senior officers reported no or lower levels of exposure to biological agents, which may be related to their experience and the roles they perform. Age also played a key role in the reported level of knowledge about the types of biological agents encountered during service, which may be influenced by professional experience. Stuart et al. conducted a study among American soldiers from the Gulf War to examine beliefs about exposure to chemical and biological weapons before and shortly after combat. The study involved 1250 male U.S. Army soldiers from the Gulf War. It was reported that 4.6% of this group believed they had been exposed to chemical and biological weapons 6 to 10 months after combat. Interestingly, individuals reporting higher levels of combat stress were at significantly greater risk of reporting that they had been exposed to chemical or biological weapons [10].

Available research indicates a decline in the incidence of infectious diseases among military personnel in recent years. One of the key factors contributing to this decline is the implementation of mandatory vaccination programs and the improvement in the quality of medical care [11]. However, increasing attention is being paid to emerging infectious diseases transmitted by mosquitoes. Social and demographic factors, such as population growth, urbanization, globalization, trade, and travel, influence this trend [12,13]. One way to reduce soldiers’ exposure to mosquito-borne diseases is through preventive and unit protection measures, such as proper use of uniforms, application of repellents to exposed skin, use of mosquito nets, and other methods [9]. These are important considerations, especially given that, according to our research, more than 75% of individuals had not participated in training on region-specific infectious diseases before being assigned to service.

Diseases related to biological risk factors also include Lyme disease, currently the most frequently diagnosed occupational disease in the general population in Poland [14]. According to our reports, as many as one in eight respondents lack knowledge about the types of biological factors to which they are exposed during service. This is significant, especially since tick bites can occur across a wide range of occupational groups, particularly among those working outdoors, including military recruits [15].

When discussing the education of soldiers, it is also essential to consider the training of military medical personnel (emergency medical services—EMS and hospitals), who frequently encounter patients with dangerous infectious diseases. Preventing the transmission of diseases in healthcare settings, including during EMS transport, involves more than just the correct use of personal protective equipment (PPE). It relies heavily on the development and implementation of appropriate administrative policies, work practices, and environmental controls, supported by targeted education, training, and supervision [16].

Although no direct evidence has been found on the effectiveness of educational programs in raising awareness among military personnel, available information suggests that education and training are crucial in preventing infectious diseases. Educational programs could potentially improve the understanding of biological threats, preventive methods, and safety procedures among military personnel, which in turn could contribute to a reduction in the incidence of infectious diseases. To confirm the effectiveness of such programs, further research focusing specifically on the impact of education on awareness and preventive behaviors of military personnel in the context of infectious diseases would be necessary.

### Limitations of the Study

The self-assessment method used in the study may be associated with potential biases related to subjective assessment of one’s own knowledge and exposure. This effect may be exacerbated by factors such as the tendency to respond in a socially desirable manner or misinterpretation of questions.

The lack of long-term tracking of both knowledge and exposure is a limitation of this study. Longitudinal studies may provide more comprehensive data on the effectiveness of educational and preventive programs.

Despite the large sample size, the results may not be fully generalizable to other military groups because military service conditions in Poland may differ from those in other countries.

## 5. Conclusions

Although the majority of respondents possess basic knowledge regarding the biological factors to which they are exposed, a significant portion is unaware of available training concerning the health effects of such exposure. Over 75% of soldiers did not participate in training related to infectious diseases specific to the region to which they were assigned. Mandatory training on local biological hazards and infectious diseases should be implemented before assignment to a specific region, as this could significantly reduce the risk of infections.

While most soldiers report feeling adequately informed about OSH rules, further efforts are needed to ensure that this information is not only available but also fully understood and applied in practice. The study results suggest that it is necessary to develop and implement health policies focused on the regular monitoring and updating of training programs to ensure they are adequate for evolving biological threats. Such policies should also take into account the needs of not only military personnel but also civilians with whom soldiers may come into contact.

Implementing these conclusions may contribute to the improvement of occupational safety and health both in military and civilian contexts.

## Figures and Tables

**Table 1 healthcare-12-01777-t001:** Self-awareness of exposure to harmful biological factors due to socio-demographic variables.

		Harmful Biological Factors				
		No Exposure [%]	Low [%]	Moderate [%]	High [%]	Very High Exposure [%]
Gender	Male	49.0	27.9	14.7	4.8	3.7
	Female	51.4	13.9	11.1	15.3	8.3
	Total	49.1	27.1	14.5	5.4	4.0
Statistics		Chi^2^ = 23.020, df = 4, *p* = 0.000				
Age	25–30 years	37.3	28.9	22.9	7.2	3.6
	31–40 years	43.6	28.6	16.6	6.3	4.9
	41–50 years	56.8	25.0	11.5	3.9	2.9
	51–60 years	54.1	28.4	8.1	6.8	2.7
	Total	49.1	27.2	14.4	5.4	3.9
Statistics		Chi^2^ = 31.101, df = 12, *p* = 0.002				
Education	Vocational	40.7	29.6	14.8	11.1	3.7
	Secondary	46.7	28.0	15.4	5.9	3.9
	Higher	52.3	25.7	13.4	4.7	4.0
	Total	49.1	27.0	14.5	5.5	3.9
Statistics		Chi^2^ = 6.44, df = 8, *p* = 0.609				
Military service corps	Professional Private	36.8	30.3	18.6	9.1	5.2
	Sergeant	49.3	26.5	14.5	5.6	4.1
	Junior Officer	57.8	26.7	11.2	0.9	3.4
	Senior Officer	66.2	23.4	8.4	0.6	1.3
	Total	49.1	27.1	14.4	5.4	4.0
Statistics		Chi^2^ = 53.586, df = 12, *p* = 0.000				
Military service tenure	3–10 years	38.9	28.0	19.8	7.8	5.4
	11–20 years	45.5	27.7	14.3	6.0	4.5
	21–30 years	62.5	24.1	9.1	2.6	1.6
	31–40 years	46.7	30.0	16.7	3.3	3.3
	>=41 years	27.8	27.8	27.8	11.1	5.6
	Total	49.1	27.0	14.5	5.5	3.9
Statistics		Chi^2^ = 49.002, df = 16, *p* = 0.000				
Military rank	Specialized—Technical, Equipment Operation, Operator, etc.	42.1	30.5	16.7	6.9	3.7
	Other Specialized	45.3	24.7	14.2	6.8	8.9
	Other	49.7	24.9	16.1	5.2	4.1
	Staff	69.7	21.4	6.7	0.8	1.3
	Other Staff Command (including Deputy, Command Assistant)	42.8	30.3	17.3	6.6	3.0
	Total	49.2	27.0	14.4	5.4	4.0
Statistics		Chi^2^ = 75.191, df = 16, *p* = 0.000				

**Table 2 healthcare-12-01777-t002:** Assessment of the level of knowledge about harmful biological factors during service due to socio-demographic variables.

		The Level of Knowledge Regarding the Types of Biological Agents Occuring During Service		
		Lack of Knowledge [%]	Partial Knowledge [%]	Comprehensive Knowledge [%]
Gender	Male	15.1	41.7	43.2
	Female	9.5	44.6	45.9
	Total	14.8	41.9	43.3
Statistics		Chi^2^ = 1.780, df = 2, *p* = 0.411		
Age	25–30 years	15.3	48.2	36.5
	31–40 years	14.2	46.2	39.6
	41–50 years	14.5	38.7	46.8
	51–60 years	21.3	26.7	52.0
	Total	14.8	42.1	43.1
Statistics		Chi^2^ = 16.709, df = 6, *p* = 0.010		
Education	Vocational	10.5	47.4	42.1
	Secondary	16.1	42.5	41.4
	Higher	13.5	41.1	45.5
	Total	14.8	41.9	43.3
Statistics		Chi^2^ = 3.206, df = 4, *p* = 0.524		
Military service corps	Professional Private	13.7	46.6	39.8
	Sergeant	15.2	41.1	43.3
	Junior Officer	16.5	43.8	39.7
	Senior Officer	13.8	33.3	52.8
	Total	14.8	42.0	43.3
Statistics		Chi^2^ = 9.952, df = 6, *p* = 0.127		
Military service tenure	3–10 years	10.8	49.6	39.6
	11–20 years	15.7	42.9	41.3
	21–30 years	15.8	36.2	48.0
	31–40 years	21.0	27.4	51.6
	Total	15.0	41.9	43.1
Statistics		Chi^2^ = 18.573, df = 6, *p* = 0.005		
Military rank	Specialized—Technical, Equipment Operation, Operator, etc.	16.4	44.9	38.6
	Other Specialized	11.6	41.4	47.0
	Other	18.3	41.1	40.6
	Staff	16.5	36.5	47.0
	Other Staff Command (including Deputy, Command Assistant)	11.3	44.3	45.4
	Total	14.9	41.8	43.3
Statistics		Chi^2^ = 12.849, df = 8, *p* = 0.117		

**Table 3 healthcare-12-01777-t003:** The level of expertise of the respondents, about trainings in the scope of the health impacts of exposure to harmful factors and prevention against them due to socio-demographic variables.

		Level of Expertise of the Respondents about Trainings in the Scope of the Health Impacts of Exposure to Harmful Factors and Prevention against Them That Were Arranged at the Workplace.			
		It Was Not Organized [%]	I Do Not Know If It Was Organized [%]	Yes, but I Did Not Participate in It [%]	Yes, I Took Part in It [%]
Gender	Male	5.3	30.6	6.4	57.6
	Female	6.8	31.1	8.1	54.1
	Total	5.4	30.7	6.5	57.4
Statistics		Chi^2^ = 0.738, df = 3, *p* = 0.864			
Age	25–30 years	5.9	38.8	11.8	43.5
	31–40 years	6.0	32.3	7.0	54.8
	41–50 years	4.3	29.3	4.8	61.6
	51–60 years	4.1	23.0	6.8	66.2
	Total	5.1	31.0	6.4	57.5
Statistics		Chi^2^ = 18.070, df = 9, *p* = 0.034			
Education	Vocational	-	21.1	36.8	42.1
	Secondary	4.9	30.1	6.1	58.9
	Higher	6.0	31.8	6.0	56.2
	Total	5.3	30.8	6.5	57.4
Statistics		Chi^2^ = 31.060, df = 6, *p* = 0.000			
Military service corps	Professional Private	6.3	30.0	9.1	54.7
	Sergeant	4.6	29.6	5.8	60.0
	Junior Officer	7.5	38.3	5.8	48.3
	Senior Officer	5.1	31.2	5.1	58.6
	Total	5.4	30.7	6.5	57.4
Statistics		Chi^2^ = 12.098, df = 9, *p* = 0.208			
Military service tenure	3–10 years	6.0	32.1	9.7	52.2
	11–20 years	5.5	32.0	6.1	56.4
	21–30 years	3.4	29.0	4.4	63.2
	31–40 years	8.1	24.2	6.5	61.3
	Total	5.2	30.9	6.4	57.5
Statistics		Chi^2^ = 15.096, df = 9, *p* = 0.088			
Military rank	Specialized—Technical, Equipment Operation, Operator, etc.	5.6	33.5	5.6	55.2
	Other Specialized	7.1	29.8	7.6	55.6
	Other	5.1	29.1	7.7	58.2
	Staff	3.6	31.0	7.1	58.4
	Other Staff Command (including Deputy, Command Assistant)	5.8	30.0	5.2	59.0
	Total	5.4	30.7	6.5	57.5
Statistics		Chi^2^ = 6.498, df = 12, *p* = 0.889			

**Table 4 healthcare-12-01777-t004:** Participation of the respondents in training in the scope of infectious diseases due to socio-demographic variables.

		Taking Part in Training in the Scope of Infectious Diseases, Specific to a Concerned Region.	
		No [%]	Yes [%]
Gender	Male	77.2	22.8
	Female	78.1	21.9
	Total	77.2	22.8
Statistics		Chi^2^ = 0.034, df = 1, *p* = 0.854	
Age	25–30 years	80.0	20.0
	31–40 years	76.1	23.9
	41–50 years	78.2	21.8
	51–60 years	78.7	21.3
	Total	77.4	22.6
Statistics		Chi^2^ = 1.147, df = 3, *p* = 0.766	
Education	Vocational	52.9	47.1
	Secondary	77.7	22.3
	Higher	77.3	22.7
	Total	77.2	22.8
Statistics		Chi^2^ = 5.784, df = 2, *p* = 0.055	
Military service corps	Professional Private	78.0	22.0
	Sergeant	77.8	22.2
	Junior Officer	75.4	24.6
	Senior Officer	74.5	25.5
	Total	77.2	22.8
Statistics		Chi^2^ = 1.117, df = 3, *p* = 0.773	
Military service tenure	3–10 years	77.0	23.0
	11–20 years	76.6	23.4
	21–30 years	79.4	20.6
	31–40 years	75.8	24.2
	Total	77.4	22.6
Statistics		Chi^2^ = 1.097, df = 3, *p* = 0.778	
Military rank	Specialized—Technical, Equipment Operation, Operator, etc.	75.4	24.6
	Other Specialized	75.8	24.2
	Other	80.8	19.2
	Staff	77.6	22.4
	Other Staff Command (including Deputy, Command Assistant)	77.1	22.9
	Total	77.2	22.8
Statistics		Chi^2^ = 2.161, df = 4, *p* = 0.706	

**Table 5 healthcare-12-01777-t005:** The respondents’ feeling of being properly informed about occupational health and safety rules at the workplace due to socio-demographic variables.

		Sense of Being Properly Informed about Occupational Health and Safety Rules at the Workplace			
		Definitely Not [%]	Rather Not [%]	Rather Yes [%]	Definitely Yes [%]
Gender	Male	0.9	3.8	50.1	45.2
	Female	-	-	50.0	50.0
	Total	0.9	3.6	50.1	45.5
Statistics		Chi^2^ = 3.814, df = 3, *p* = 0.282			
Age	25–30 years	-	2.4	57.6	40.0
	31–40 years	1.0	3.7	55.6	39.7
	41–50 years	0.8	3.1	45.7	50.5
	51–60 years	-	5.3	34.7	60.0
	Total	0.8	3.5	50.4	45.3
Statistics		Chi^2^ = 24.331, df = 9, *p* = 0.004			
Education	Vocational	-	18.2	50.0	31.8
	Secondary	0.9	2.7	49.9	46.5
	Higher	0.9	4.4	50.3	44.4
	Total	0.8	3.8	50.1	45.3
Statistics		Chi^2^ = 16.162, df = 6, *p* = 0.013			
Military service corps	Professional Private	0.9	3.1	50.9	45.0
	Sergeant	1.2	3.2	50.3	45.4
	Junior Officer	-	3.3	57.9	38.8
	Senior Officer	-	6.3	42.5	51.3
	Total	0.8	3.6	50.2	45.4
Statistics		Chi^2^ = 12.628, df = 9, *p* = 0.180			
Military service tenure	3–10 years	0.7	2.6	54.5	42.2
	11–20 years	0.8	3.7	53.5	42.2
	21–30 years	0.9	3.4	44.3	51.4
	31–40 years	-	4.8	33.9	61.3
	Total	0.8	3.4	50.3	45.5
Statistics		Chi^2^ = 17.240, df = 9, *p* = 0.045			
Military rank	Specialized—Technical, Equipment Operation, Operator, etc.	-	4.4	54.4	41.2
	Other Specialized	2.0	2.5	49.5	46.0
	Other	1.5	3.6	43.7	51.3
	Staff	0.4	5.3	50.4	44.0
	Other Staff Command (including Deputy, Command Assistant)	0.8	2.2	51.0	46.0
	Total	0.9	3.6	50.2	45.4
Statistics		Chi^2^ = 12.628, df = 12, *p* = 0.121			

## Data Availability

All data generated or analyzed during this study are included in this published article.

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
