# Peer review of "Evaluation of Soldiers’ Knowledge and Sense of Threats Regarding Exposure to Biological Risk Factors at the Place of Service"

_healthcare, 2024, doi:10.3390/healthcare12171777_

Round 1

Reviewer 1 Report

Comments and Suggestions for Authors

The article is another presentation from a series of studies on a very important issue, which is knowledge about the occurrence of harmful factors occurring in connection with military service. The previous article concerned knowledge about physical and chemical factors, the current one concerns knowledge about exposure to biological factors at the place of service. The studies were well planned and conducted by a facility specialized in social research in the Polish Army. The specific research objectives concerned knowledge about the level of exposure to harmful biological factors, the knowledge regarding the types of biological agents and the level of expertise of the respondents about training in the scope of the health impacts of exposure to harmful factors and prevention against them, that were arranged at the workplace. The authors checked whether and to what extent the answers to the questions posed depend on socio-demographic variables of the respondents. However, in the discussion they did not refer in any way to the results of this tedious analysis. Moreover, if the question is asked about the level of knowledge about a factor, it is worth checking whether participation in appropriate training had an impact on it. This will allow for the improvement of training. To sum up, in my opinion the article requires the indicated minor corrections.

Author Response

Thank you very much for your valuable suggestions.

I greatly appreciate the constructive comments. The comments were useful in improving the manuscript and increased the scientific value of the revised manuscript.

(…) However, in the discussion they did not refer in any way to the results of this tedious analysis. Moreover, if the question is asked about the level of knowledge about a factor, it is worth checking whether participation in appropriate training had an impact on it. This will allow for the improvement of training. To sum up, in my opinion the article requires the indicated minor corrections.

The comments were applied.

Reviewer 2 Report

Comments and Suggestions for Authors

The manuscript reported the KAP of military soldiers on biological risk factors at their service place. It is a good topic, but the quality of both study design and manuscript writing was low. A major revision of the manuscript is required.

Firstly, the major content of the questionnaire used in this survey should be described since it is self-designed. The meaning of different grading should be explained, such as exposure degrees (No, Low, Moderate, High, and Very High); levels of knowledge (Lack, Partial, and Comprehensive); Sense of being properly informed (Definitely not, Rather not, Rather ye, Definitely yes).

Secondly, the results should be presented in a simplified form, the tables need to be reformed as a standardized requirement. Definitely, which biological factors the soldiers are exposed to in their service places should be reported since preventive measures can vary greatly against different biological factors depending on their harmfulness.

Thirdly, the abstract should be rewritten, the key information should be included. The current abstract presented too much background information, and no information on how the authors conducted the survey, and the major findings.

Besides, the data introduced in the section of the 2.2. Characteristics of the respondents can be simplified. It was basic information about the study subjects and can be put into the result section.

Finally, the limitations of this study should be discussed, for example, there must be bias in sampling choice.

Author Response

Thank you very much for your valuable suggestions.

I greatly appreciate the critical and constructive comments. The comments were useful in improving the manuscript and increased the scientific value of the revised manuscript. The manuscript has been revised according to the comments

  1. The manuscript reported the KAP of military soldiers on biological risk factors at their service place. It is a good topic, but the quality of both study design and manuscript writing was low. A major revision of the manuscript is required.

The manuscript has been thoroughly revised.

  1. Firstly, the major content of the questionnaire used in this survey should be described since it is self-designed. The meaning of different grading should be explained, such as exposure degrees (No, Low, Moderate, High, and Very High); levels of knowledge (Lack, Partial, and Comprehensive); Sense of being properly informed (Definitely not, Rather not, Rather ye, Definitely yes).

Thank you for your advice. Your comments have been taken into account and necessary corrections have been made.

  1. Secondly, the results should be presented in a simplified form, the tables need to be reformed as a standardized requirement. Definitely, which biological factors the soldiers are exposed to in their service places should be reported since preventive measures can vary greatly against different biological factors depending on their harmfulness.

The form of reporting the results is compatible with the first part of the results published in Healthcare, 2024, vol. 12, no. 4, pp. 1-12. The aim of the entire study, a fragment of which is described in this manuscript, was to examine general knowledge and self-awareness of exposure to broadly understood biological factors, without specifically indicating them.

  1. Thirdly, the abstract should be rewritten, the key information should be included. The current abstract presented too much background information, and no information on how the authors conducted the survey, and the major findings.

The comments were applied.

  1. Besides, the data introduced in the section of the 2.2. Characteristics of the respondents can be simplified. It was basic information about the study subjects and can be put into the result section.

Comments were taken into account. The subsection: Characteristics of respondents was simplified. However, as this manuscript is a continuation of the work described in Healthcare, 2024, vol. 12, no. 4, pp. 1-12, it was left in the section: Material and methods.

  1. Finally, the limitations of this study should be discussed, for example, there must be bias in sampling choice.

The comments were taken into account. The limitations of the study were included in the manuscript.

Reviewer 3 Report

Comments and Suggestions for Authors

Overall Comments:

Interesting topic with important implications. Please re-read each section for better flow, transitions, grammatical/syntax, tenses (back and forth between present and past), typos, and other English language edits.

Abstract:

Comment 1:

Suggest slightly revising the abstract-especially the second half-for improved comprehension.

Introduction:

Comment 1:

While bullet points in parts of the text can be acceptable, the inclusion of such in the introduction cuts the flow. Suggest incorporating these important elements in a well-crafted paragraph.

Comment 2:

The authors may want to clarify a bit more the difference between the focus here (i.e. biological risk factors) and physical and chemical risk factors, as well as intended/warfare/weapons exposure versus unintended exposure (e.g., mosquitoe bite).

Comment 3:

The intended population is military personnel. However, as the authors point out, given the possibility of transmission of biological exposure to civilians at large and the widespread implications of this, the authors might consider emphasizing the importance of the latter here and later on in the conclusion.

Materials and Methods:

Comment 1:

The reader only learns and assumes that the sample is of Polish military personnel when the committee name is listed. Consider indicating this in the opening sentence.

Comment 2:

Suggest incorporating the hypotheses in paragraph form rather than as such as, respectfully, it reads more like a graduate-level paper/dissertation than an academic article.

Comment 3:

Please carefully re-read and edit this paragraph: «Gender, age, place of residence of the respondents

Comment 4:

Bullet points are not needed in this section.

Comment 5:

Consider streamlining redundancy between sub-titles of paragraphs, introductory sentences of paragraphs, and table titles.

Comment 6:

Finally, if the authors have access to other data, these questions come to mind:

1. Any distinctions in results/findings by military faction?

2. Since nationwide, any differences in knowledge or outcomes depending on geographic location?

3. Any differences in occupational safety and health knowledge when looking at biological risk factors versus physical or chemical risk factors?

4. Were the authors able to ascertain what nationwide and/or local mandatory occupational safety and health training/policies surrounding these issues are in place and if these are being rolled out consistently and comprehensively by those in charge?

Discussion:

Comment 1:

While there are some interesting points in the discussion, some of these ought to be moved to the introduction and others should be removed altogether. Suggest carefully revising this section to address this as well as to re-order some elements. A limitations and strengths section would also be appropriate here.

Conclusion:

Comment 1:

Again, I do not see why bullet points are included here. A simple paragraph would read better.

Comment 2:

Additionally, not only are included bullet points redundant to what is earlier provided in the results and/or discussion section, but these are not conclusions. Aside from the final sentence, this section does not provide the kind of take-away points that could prove useful. Strongly suggest writing up a concise section that readers can use to inform future research, practice and policy surrounding occupational safety and health for military personnel, which could also impact non-military personnel with whom they may come in contact.

Comments on the Quality of English Language

Please re-read each section for better flow, transitions, grammatical/syntax, tenses (back and forth between present and past), typos, and other English language edits.

Author Response

Thank you very much for your valuable suggestions.

I greatly appreciate the critical and constructive comments. The comments were useful in improving the manuscript and increased the scientific value of the revised manuscript. The manuscript has been revised according to the comments

Interesting topic with important implications. Please re-read each section for better flow, transitions, grammatical/syntax, tenses (back and forth between present and past), typos, and other English language edits.

 Abstract:

Comment 1:

Suggest slightly revising the abstract-especially the second half-for improved comprehension.

 The comments were applied.

Introduction:

Comment 1:

While bullet points in parts of the text can be acceptable, the inclusion of such in the introduction cuts the flow. Suggest incorporating these important elements in a well-crafted paragraph.

 The comments were applied.

Comment 2:

The authors may want to clarify a bit more the difference between the focus here (i.e. biological risk factors) and physical and chemical risk factors, as well as intended/warfare/weapons exposure versus unintended exposure (e.g., mosquitoe bite).

 The comments were applied.

Comment 3:

The intended population is military personnel. However, as the authors point out, given the possibility of transmission of biological exposure to civilians at large and the widespread implications of this, the authors might consider emphasizing the importance of the latter here and later on in the conclusion.

 The comments were applied.

Materials and Methods:

Comment 1:

The reader only learns and assumes that the sample is of Polish military personnel when the committee name is listed. Consider indicating this in the opening sentence.

 The comments were applied.

Comment 2:

Suggest incorporating the hypotheses in paragraph form rather than as such as, respectfully, it reads more like a graduate-level paper/dissertation than an academic article.

 The comments were applied.

Comment 3:

Please carefully re-read and edit this paragraph: «Gender, age, place of residence of the respondents.»

 The comments were applied.

Comment 4:

Bullet points are not needed in this section.

 The comments were applied.

Comment 5:

Consider streamlining redundancy between sub-titles of paragraphs, introductory sentences of paragraphs, and table titles.

Comments have been implemented. Redundancy has been improved in the Results section.

Comment 6:

Finally, if the authors have access to other data, these questions come to mind:

  1. Any distinctions in results/findings by military faction?

Thank you for your comment. It will be used in planning the next research. Unfortunately, we do not have the data due to the military faction.

  1. Since nationwide, any differences in knowledge or outcomes depending on geographic location?

Thank you for your comment. Geographic location due to the specifics of a soldier's work and their relocation often several times a year, in our opinion is not a significant demographic variable.

  1. Any differences in occupational safety and health knowledge when looking at biological risk factors versus physical or chemical risk factors?

Thank you for your comment. Comparison of biological factors with chemical and physical factors was not part of this manuscript, but may be the subject of a future paper.

  1. Were the authors able to ascertain what nationwide and/or local mandatory occupational safety and health training/policies surrounding these issues are in place and if these are being rolled out consistently and comprehensively by those in charge?

From the knowledge we have, occupational health and safety training is organized, however, as the results of this study show, it is insufficient and information about it does not reach all interested parties.

The work aimed to examine the general knowledge and self-awareness of soldiers in order to implement training in this area in the future.

Discussion:

Comment 1:

While there are some interesting points in the discussion, some of these ought to be moved to the introduction and others should be removed altogether. Suggest carefully revising this section to address this as well as to re-order some elements. A limitations and strengths section would also be appropriate here.

 The comments were applied.

Conclusion:

Comment 1:

Again, I do not see why bullet points are included here. A simple paragraph would read better.

 The comments were applied.

Comment 2:

Additionally, not only are included bullet points redundant to what is earlier provided in the results and/or discussion section, but these are not conclusions. Aside from the final sentence, this section does not provide the kind of take-away points that could prove useful. Strongly suggest writing up a concise section that readers can use to inform future research, practice and policy surrounding occupational safety and health for military personnel, which could also impact non-military personnel with whom they may come in contact.

The comments were applied.

Round 2

Reviewer 2 Report

Comments and Suggestions for Authors

No specific comment or suggestion.